# Quinoxaline-based anti-schistosomal compounds have potent anti-plasmodial activity

Mukul Rawat[1,2], Gilda Padalino[3,4], Edem Adika[1], John Okombo[5,6], Tomas Yeo[5,6], Andrea Brancale[7], David A. Fidock[5,6], Karl F. Hoffmann[3], Marcus C. S. Lee[1,2]*

1 Biological Chemistry and Drug Discovery, Wellcome Centre for Anti-Infectives Research, University of Dundee, Dundee, United Kingdom, 2 Wellcome Sanger Institute, Wellcome Genome Campus, Hinxton, United Kingdom, 3 Department of Life Sciences (DLS), Aberystwyth University, Aberystwyth, United Kingdom, 4 Swansea University Medical School, Swansea, United Kingdom, 5 Department of Microbiology and Immunology, Columbia University Irving Medical Center, New York, New York, United States of America, 6 Center for Malaria Therapeutics and Antimicrobial Resistance, Division of Infectious Diseases, Department of Medicine, Columbia University Irving Medical Center, New York, New York, United States of America, 7 Department of Organic Chemistry, University of Chemistry and Technology Prague, Prague, Czech Republic

* mlee001@dundee.ac.uk

## Abstract

The human pathogens *Plasmodium* and *Schistosoma* are each responsible for over 200 million infections annually, especially in low- and middle-income countries. There is a pressing need for new drug targets for these diseases, driven by emergence of drug-resistance in *Plasmodium* and an overall dearth of drug targets against *Schistosoma*. Here, we explored the opportunity for pathogen-hopping by evaluating a series of quinoxaline-based anti-schistosomal compounds for their activity against *P. falciparum*. We identified compounds with low nanomolar potency against 3D7 and multidrug-resistant strains. *In vitro* resistance selections using wildtype and mutator *P. falciparum* lines revealed a low propensity for resistance. Only one of the series, compound **22**, yielded resistance mutations, including point mutations in a non-essential putative hydrolase *pfqrp1,* as well as copy number amplification of a phospholipid-translocating ATPase, *pfatp2*, a potential target. Notably, independently generated CRISPR-edited mutants in *pfqrp1* also showed resistance to compound **22** and a related analogue. Moreover, previous lines with *pfatp2* copy number variations were similarly less susceptible to challenge with the new compounds. Finally, we examined whether the predicted hydrolase activity of PfQRP1 underlies its mechanism of resistance, showing that both mutation of the putative catalytic triad and a more severe loss of function mutation elicited resistance. Collectively, we describe a compound series with potent activity against two important pathogens and their potential target in *P. falciparum*.

## Author summary

The emergence of resistance to front line antimalarial therapies has been a recurring problem for malaria control efforts, and new treatments would ideally have a low risk of resistance and modest loss of potency should resistance arise. Here, we explored the

**Data availability statement:** All associated sequence data are available at the European Nucleotide Archive under accession code PRJEB74174. https://www.ebi.ac.uk/ena/browser/view/PRJEB74174.

**Funding:** This work was supported by funding from Wellcome (206194/Z/17/Z; wellcome.org) to MCSL, from the National Institutes of Health, National Institute of Allergy and Infectious Diseases (R01 AI124678, R01 AI185559 and R01 AI109023; www.nih.gov/grants-funding) and the Bill & Melinda Gates Foundation (INV-033538; www.gatesfoundation.org) to DAF, and from the Life Sciences Research Network Wales (www.lsrnw.ac.uk/home) to KFH and AB. The funders had no role in study design, data collection and analysis, decision to publish, or preparation of the manuscript.

**Competing interests:** The authors have declared that no competing interests exist.

mode of action and resistance risk against the malaria parasite *Plasmodium falciparum* of a chemical series of quinoxaline-based compounds originally developed against another blood-feeding parasite, the blood fluke *Schistosoma*. A subset of compounds were highly potent against *P. falciparum* strains, and were challenging to elicit *in vitro* resistance. Whole-genome sequence analysis of resistant parasites identified point mutations in a putative membrane hydrolase. We show that engineered loss-of-function mutants of the hydrolase confer resistance, suggesting this as a resistance mechanism rather than a drug target. In addition, in the drug-pressured lines we observed a copy number amplification of a gene encoding a phospholipid flippase, a potential target and class of transporter involved in maintaining membrane asymmetry. Collectively, our study shows that quinoxaline-based compounds are potent, have low resistance risk, and act via a potentially novel mode of action against malaria parasites.

## Introduction

Although significant progress has been made in malaria elimination, there were an estimated 263 million new cases and 597,000 deaths due to malaria infection in 2023 [1]. Artemisinin remains the gold standard antimalarial for uncomplicated malaria, with artemisinin-based combination therapies the recommended treatment since 2005. These combinations comprise fast-acting yet short-lived artemisinin derivatives that significantly reduce initial parasite biomass, partnered with longer-acting molecules with distinct modes of action to eliminate residual parasites. The emergence of resistance to artemisinin and partner drugs in Southeast Asia, and more recently artemisinin partial resistance in East Africa are severe threats to malaria control and elimination [2,3]. Delaying the emergence of resistance will require the identification of new anti-plasmodial modes of action, ideally with low propensity for resistance.

To guide antimalarial drug discovery, the Medicines for Malaria Venture (MMV) proposed desired types of molecules (Target Candidate Profiles) and medicines (Target Product Profiles) [4]. Decades of research and discovery have led to diverse molecules in preclinical and early clinical stages. A common route for antimalarial drug discovery is phenotypic screening, where distinct parasite stages are co-incubated with compounds to identify active molecules. While targets are not necessarily known at this stage in the process, approaches such as the *in vitro* evolution of resistant parasites using low drug pressure ranging from sub-lethal to $3 \times IC_{90}$, followed by whole-genome sequencing to identify newly-generated mutations, can be used to deconvolute targets as well as compound mode of action [5,6].

Quinoxaline derivates are a class of heterocyclic compounds, characterised by a fusion of a benzene and a pyrazine ring, which have diverse applications in medicine due to their biological activities. In addition to their known anti-microbial, anti-inflammatory, anti-cancer, anti-depressant, and anti-diabetic activities [7], quinoxaline derivatives also demonstrate anti-plasmodial properties. A Novartis chemical library screen identified BQR695 (2-[[7-(3,4-dimethoxyphenyl)quinoxalin-2-yl]amino]-N-methylacetamide) as an anti-plasmodial compound that acts through inhibition of phosphatidylinositol 4-kinase (PfPI4K) [8]. More recently, we showed that quinoxaline compounds possess a low potential for resistance, requiring the use of a mutator parasite line to evolve even low-level resistance [9].

*Plasmodium* and *Schistosoma* are hematophagous species that ingest and degrade host hemoglobin for nutrition and share a common mechanism of dispensing with the toxic 'free' heme molecules liberated from this process by sequestering them into inert hemozoin [10]. As part of anti-plasmodial and anti-schistosomal drug discovery efforts, inhibition of

this pathway in both organisms has been demonstrated using various chemical scaffolds [11,12]. In this study, we explored a pathogen-hopping opportunity, evaluating a series of quinoxaline-containing compounds that have been previously shown to have anti-schistosomal activity [13]. A subset of these compounds have highly potent anti-plasmodial activity, with single-digit nanomolar $IC_{50}$ values. Using *in vitro* resistance evolution experiments to gain insights into the mode of compound action, we identified mutations in quinoxaline resistance protein (PfQRP1), a non-essential putative hydrolase. In addition, we showed that parasites with copy number amplification of a phospholipid-translocating ATPase, *pfatp2*, were also less susceptible to these quinoxaline-based compounds. Leveraging a two-part protocol that tests for inhibition of hemozoin formation, we provide evidence of possible involvement of quinoxaline analogs in perturbation of hemoglobin breakdown but not downstream hemozoin formation. Overall, we show that quinoxaline-like compounds can be potent anti-infectives, with a low propensity for resistance and a modest loss of potency in resistant *P. falciparum* parasites. The dual activity against both *Plasmodium* and *Schistosoma* hints at a conserved target or pathway, suggesting that further exploration of these compounds in *Plasmodium* may also yield insights for accelerating anti-schistosomal drug discovery.

## Results

### Activity of quinoxaline compounds against *Plasmodium falciparum*

We investigated the anti-plasmodial activity of a lead anti-schistosomal molecule, compound **22** [13] (Fig 1A). This compound showed potent activity against both 3D7 ($IC_{50}$ = 22 nM) and the multi-drug resistant strain Dd2 ($IC_{50}$ = 32 nM). Modification of the nitro group on the C6 position of the central core to either a *N*-acetyl amide (compound **22c**) or *N*-furan-2-carboxamide (compound **22f**) greatly diminished activity against both *P. falciparum* strains (Fig 1A), similar to the effect on anti-schistosomal activity [13].

Keeping the C6 nitro group constant, we further explored 8 additional derivatives with modifications of the aromatic rings and the *N*-linker between the quinoxaline core and each aromatic ring (S1 Fig and S1 Table). Introduction of a *N*-ethyl linker (compound **37**) resulted in a ten-fold loss in potency (Fig 1B). In contrast, modifications on the aromatic rings strongly influenced phenotypic activity. In fact, the introduction of a trifluoromethyl group in compounds **30–33** led to an increase in potency, with single-digit nanomolar $IC_{50}$ values for compounds **31** and **33** (Fig 1B and S2 Table).

We further evaluated compounds **22**, **31** and **33** against a panel of geographically distinct parasite strains, including two Cambodian isolates with multi-drug resistance (to artemisinin, chloroquine, and pyrimethamine), as well as parasites from Uganda (UG659), Ghana (GB4) and Tanzania. All of the strains were affected by compound treatment with potencies similar to those recorded for the lab strain Dd2 (Fig 1C). To assess whether the compounds showed stage-specific activity, tightly synchronised parasites in the ring, trophozoite and schizont stages were pulsed with compounds **22**, **31** and **33** for 10 h, followed by washout and continued culture for a total of 72 h. The compounds did not demonstrate a strong stage-specificity, and overall showed only a 3–4 fold drop in potency for a 10 h pulse compared with continuous exposure for 72 h (S2A–S2C Fig). To examine the rate of parasite killing after compound treatment, we treated cultures with compounds **22**, **31** and **33** for 24 h or 48 h, and measured invasion of remaining viable parasites into red blood cells (RBCs) pre-labelled with CFDA-SE (carboxylfluorescein diacetate succinimidyl ester [14]). Compounds **22**, **31** and **33** rapidly reduced parasitemia *in vitro* at a rate only modestly slower than fast-acting drugs dihydroartemisinin and chloroquine, and more rapidly than the slow-acting drug atovaquone (S3 Fig).

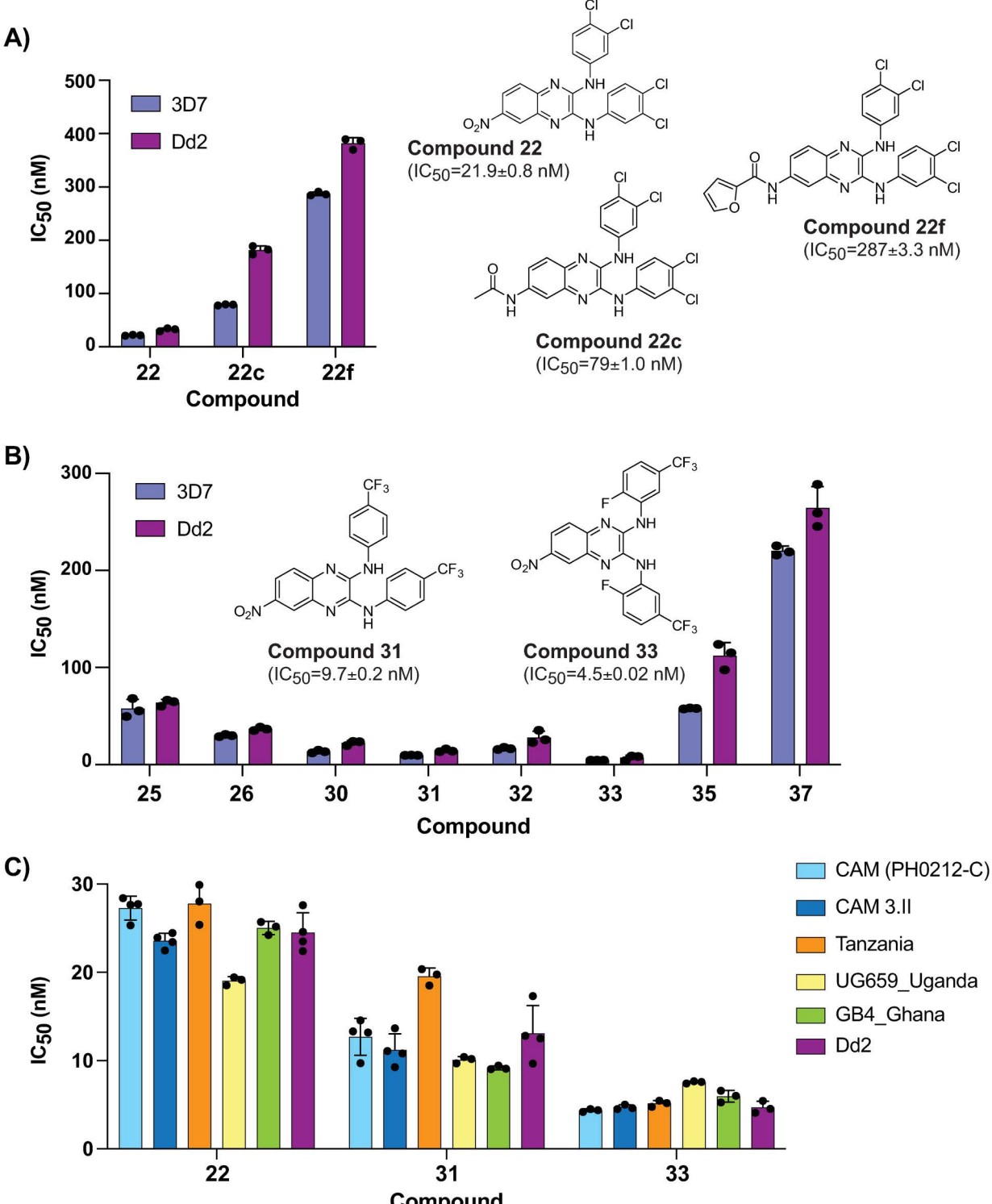

**Fig 1. Anti-schistosomal compounds show potent anti-plasmodial activity.** (A) The lead compound **22** and two derivatives show a range of activities against the *P. falciparum* strains 3D7 and Dd2. Structures of all compounds are in S1 Fig, and selected structures are shown with 3D7 $IC_{50}$ values. **(B)** Evaluation of additional derivatives of compound **22** identified compounds **31** and **33** as the most potent. **(C)** Anti-plasmodial activity of compounds **22**, **31**, and **33** is comparable against a panel of geographically diverse strains. Values for all compounds shown in S1 Table. For (A–C) each dot represents a biological replicate (n = 3–4) with mean±SD values shown as a bar chart.

Overall, comparison of the activity between *P. falciparum* asexual blood stage parasites and *S. mansoni* schistosomula showed a high correlation across all 11 compounds tested (S4A Fig and S2 Table). In contrast, previously reported cytotoxicities of these compounds against HepG2 cells [13] showed a low correlation with *P. falciparum* 3D7 potency (S4A Fig) and a moderate selectivity index range of 3 to 67 (S4B Fig). We next investigated whether the compounds share chemical features (e.g., Lipinski's rule of five (Ro5) [15] and absorption, distribution, metabolism and excretion (ADME) metrics [16]) similar to those found in orally delivered drugs. The Ro5 analysis, using SwissADME [17] calculated each compound's molecular weight (MW), calculated Log P (cLogP), number of hydrogen bond donors (HBD), number of hydrogen bond acceptors (HBA), and number of rotatable bonds (NROTB). Except for three compounds containing two Ro5 violations (S3 Table, compounds **22f**, **32** and **33**), all other compounds were within the acceptable pharmacokinetic and physiochemical ranges set for each descriptor (≤ 1 violation). By comparing to recognised guidelines [18], the majority of each compound's predicted ADME parameters were within the recommended ranges recognised for each parameter (S4 Table). The one exception is the potential for these compounds to inhibit human ether-a-go-go-related gene (hERG) K+ channel activity, which may cause drug-induced severe cardiac side-effects. In summary, the Ro5 and ADME analyses predicted that these compounds generally have good chemical properties for an orally delivered therapeutic. However, future efforts should focus on developing drug-like compounds with a reduced potential to inhibit hERG activity.

## Resistance generation using a mutator parasite

*In vitro* evolution of resistance is a powerful tool for understanding drug mode of action [5]. To facilitate the isolation of mutants, we recently reported a mutator *P. falciparum* line that has an elevated mutation rate and higher propensity to select for resistance, resulting from defective proof-reading due to two introduced mutations in the DNA polymerase δ catalytic subunit [9]. To investigate the mode of action of these quinoxaline compounds, the Dd2-Polδ mutator line was pressured with compounds **22**, **31** and **33** (Fig 2A). A single-step *in vitro* resistance evolution method was used where triplicate flasks with $1 \times 10^8$ parasites were exposed to $5 \times IC_{50}$ of each compound (Fig 2A). After 8–10 days of treatment, parasites were undetectable (<0.1% parasitemia) by microscopy for all three compounds. Drug pressure was then removed, and parasites were allowed to recover. After approximately 3 weeks, compound **22**- and **31**-treated parasites recovered in two of the triplicate flasks tested. In contrast compound **33**-treated parasites did not recover even after 60 days.

Clonal lines were isolated using limiting dilution of bulk cultures from parasites treated with compounds **22** and **31**. Clones derived from compound-**22** selections from flask 1 (C22-1.1) and flask 2 (C22-2.1) showed a small but consistent shift in $IC_{50}$ to both compounds **22** and **31** (Fig 2B), but unexpectedly not to compound **33** (S5 Fig). To explore this further, we examined structural isomers of compound **33** and compound **31** (compound **32** and **30**, respectively), as well as compound **35**, which differs from compound **30** by a methoxy group in the *meta* position on both *N*-aryl groups. Apart from compound **33**, each of these compounds showed a modest shift with one of the compound **22**-resistant clones (S5 Fig).

In contrast to the compound **22** resistance selections, cultures pressured with compound **31** showed no significant shift in $IC_{50}$ compared to the parental line. To explore if we could generate resistance to compounds **31** or **33**, we repeated the resistance selections using $1 \times 10^9$ parasites as the initial inoculum, however, no recrudescent parasites were recovered (Fig 2A). Ramping selections where parasites were cultured initially at $1 \times IC_{50}$ and slowly adapted to increasing concentrations of drug also yielded no resistance, with parasites unable to proliferate at $2 \times IC_{50}$ despite exposing them for almost a month. Thus, the resistance risk with these

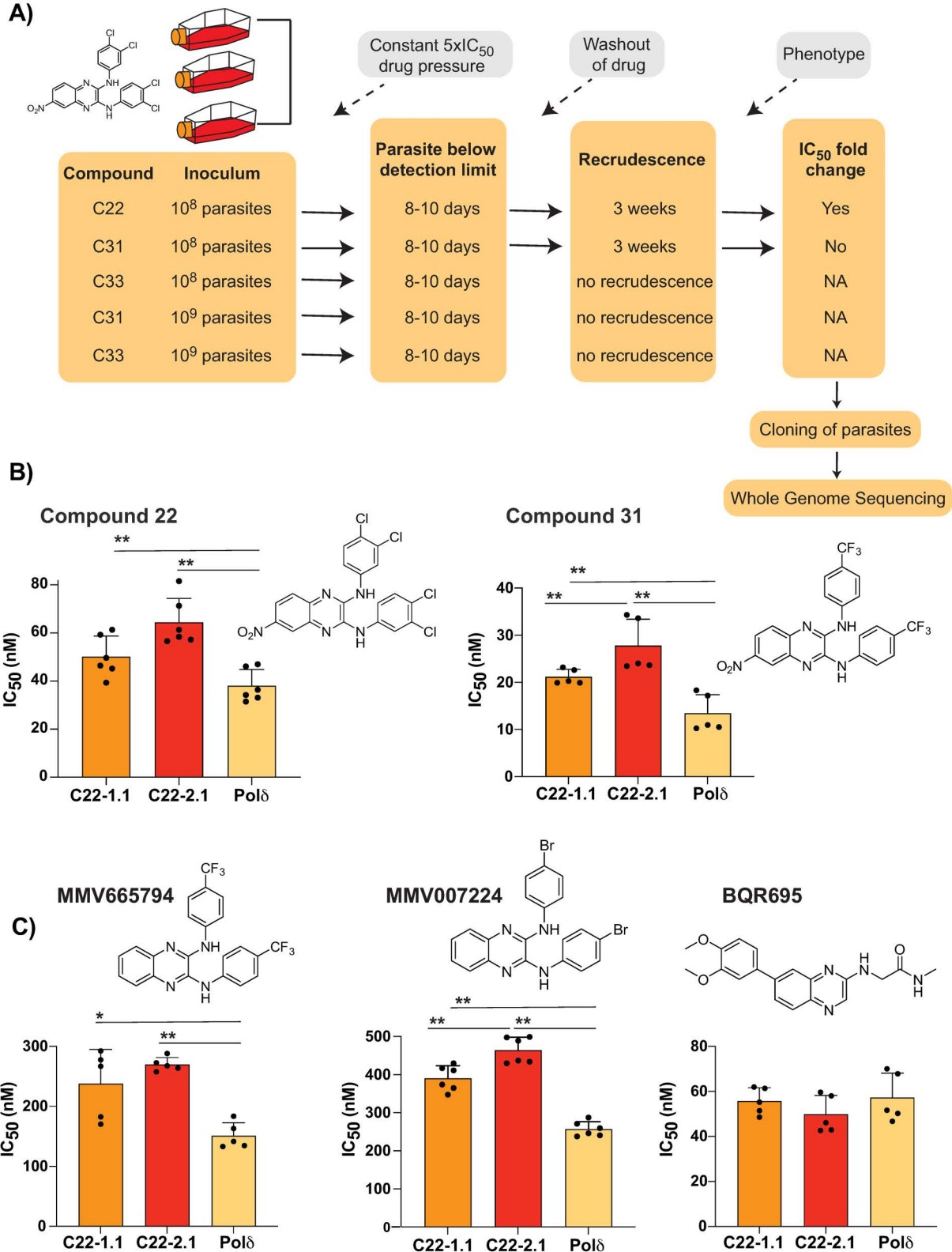

**Fig 2. Evolution of resistance to quinoxaline-containing compounds. (A)** Selection scheme showing attempts to generate resistance using the Dd2-Polδ mutator line. Indicated are the compound, parasite inoculum, selection period during which parasites were exposed to 5×IC$_{50}$, recrudescence status of cultures, and shift in IC$_{50}$ of the bulk culture. **(B)** Clones from two independent flasks (1.1 and 2.1) from the compound

22 selection were evaluated against compounds **22** and **31**. The parental line, Dd2-Polδ (Polδ) was used as the control. **(C)** Clones (1.1 and 2.1) from the compound **22** selection were evaluated against other quinoxaline-containing compounds, MMV665794, MMV007224 and BQR695. Each dot represents a biological replicate (n = 4–6) with mean±SD shown as bar chart, and statistical significance determined by Mann-Whitney $U$ test (*p < 0.05, **p < 0.01).

quinoxaline-based analogues was low, with compounds **31** and **33** proving to be resistance-refractory to date and compound **22** yielding only low-level resistance.

To more formally establish a minimum inoculum for resistance (MIR; [6,19]) estimation for compound **22**, we performed resistance selections with both parental Dd2 and the isogenic Dd2-Polδ mutator parasite using triplicate cultures at an inoculum of $10^4 – 10^8$ parasites per culture. As a control, we pressured parasites with the compound GNF179, which is related to the clinical candidate ganaplacide and for which resistance would be expected at ~$10^7$ for wildtype parasites [20]. Recrudescence was observed for GNF179 in 3/3 cultures at $10^7$ and $10^8$ for the parental Dd2 line, and 2 of 3 cultures at the lower inoculum of $10^6$ for the Dd2-Polδ mutator (S6A Fig). In contrast, no recrudescence was observed for compound **22** selections with either line (S6B Fig). For compound **22** the MIR was therefore >$3×10^8$ parasites

The low propensity for resistance of these quinoxaline compounds was reminiscent of our experience with *in vitro* evolution experiments with two related quinoxaline scaffolds (2,3-dianilinoquinoxaline derivatives without nitro group on the C6 position): MMV665794 (2-N,3-N-bis[3-(trifluoromethyl)phenyl]quinoxaline-2,3-diamine) and MMV007224 (2-*N*,3-*N*-bis(4-bromophenyl)quinoxaline-2,3-diamine) [9,21]. Examination of cross-resistance of the compound **22**-selected clones to these two compounds revealed a similar low-level shift in $IC_{50}$, suggesting a shared mechanism (Fig 2C). In contrast, no cross resistance was found against the PfPI4K inhibitor BQR695 [8], which possesses a quinoxaline core but not the 2,3-dianilino substitution pattern (Fig 2C).

## Mutations in PfQRP1 confer resistance

To identify mutations in the compound **22**-selected parasites, we performed whole-genome sequencing of eight clones isolated from two separate flasks (clones 1.1–1.4 and clones 2.1–2.4) as well as the sensitive isogenic parent. Only one gene, PF3D7_1359900, had unique mutations relative to the parent in all clones, with seven clones encoding a R676I mutation and another encoding a V673D mutation (Fig 3A and S5 Table). Notably, this gene was also mutated in previous selections with the compound MMV665794 described above, and encodes a protein we recently designated PfQRP1, for quinoxaline resistance protein [9].

Although the function of PfQRP1 is unknown, the 2126 amino acid protein is predicted to encode four transmembrane domains spanning residues 412–530, and a putative alpha-beta hydrolase domain at the C-terminus (Fig 3A). The sites of the compound **22**-resistance mutations, V673D and R676I, are relatively well conserved across orthologous Apicomplexan proteins (Fig 3B). When mapped onto an AlphaFold-generated structure of PfQRP1, these residues as well as the two previously identified mutations G1612V and D1863Y that confer resistance to MMV665794 [9], are located close to the putative catalytic triad of the hydrolase domain (Fig 3C).

To validate the importance of PfQRP1 for resistance to the anti-schistosomal compounds **22** and **31**, we next used previously generated CRISPR-edited mutant lines bearing the G1612V and D1863Y mutations, as well as control edited lines with only silent mutations at those sites. Both compound **22** and **31** showed reduced susceptibility in both CRISPR-edited mutant lines (Fig 3D–3F).

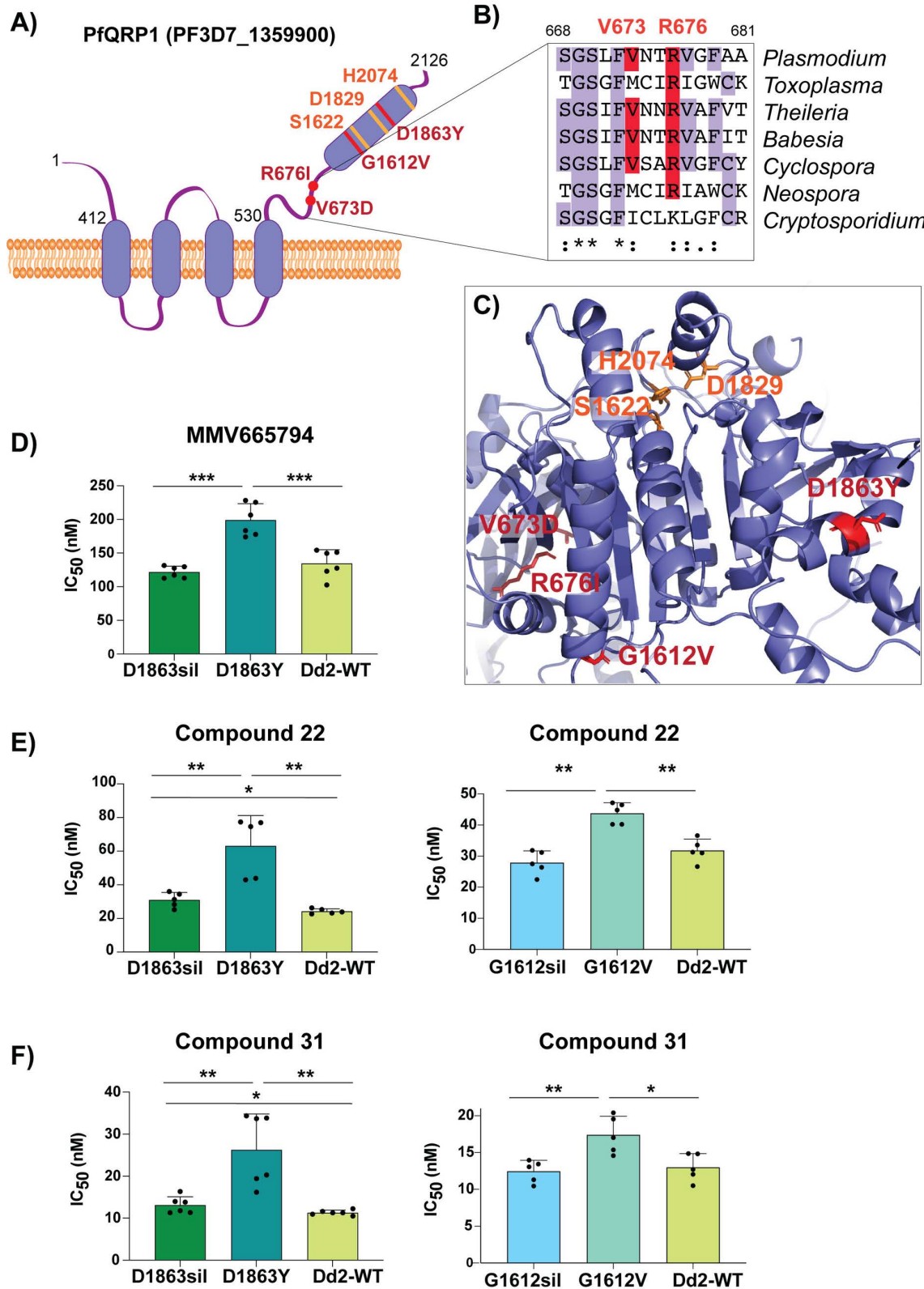

**Fig 3. Mutations in PfQRP1 confer resistance to quinoxaline-containing compounds. (A)** Model of the PfQRP1 protein, with 4 predicted transmembrane segments and a putative α/β hydrolase domain. The compound **22**-resistance mutations V673D and R676I, and the MMV665794-resistance mutations G1612V and D1863Y are shown in red, and putative catalytic triad in yellow. **(B)**

Partial sequence alignment of *pfqrp1* homologs showing partial conservation of residues mutated in drug-selected parasites (red). *P. falciparum pfqrp1* (PF3D7_1359900) with *Toxoplasma gondii* (TGME49_289880), *Theileria parva* (TpMuguga_02g02080), *Babesia bovis* (BBOV_II004840), *Neospora caninum* (NCLIV_042110), *Cyclospora cayetanensis* (cyc_01400) and *Cryptosporidium parvum* (cgd3_590) orthologs. **(C)** AlphaFold model of PfQRP1 showing the resistance mutations (red) and the residues of the putative catalytic triad (yellow). **(D–F)** CRISPR-edited parasite lines with the MMV665794-resistance mutations D1863Y and G1612V show a shift against MMV665794 **(D)** as well as compounds **22 (E)** and **31 (F)**. Control lines with only silent mutations (sil) or the unedited wildtype (WT) are shown. Each dot represents a biological replicate (n = 5–6) with mean±SD shown as bar chart, and statistical significance determined by Mann-Whitney *U* tests (*p < 0.05, **p < 0.01).

## Amplification of a lipid flippase confers resistance to quinoxaline compounds

Our observations above demonstrate the importance of PfQRP1 as a resistance mechanism to these quinoxaline-based compounds. However, PfQRP1 is unlikely to be the target, as the gene is non-essential based on mutagenesis in the *piggyBac* screen, as well as the presence of a frameshift mutation near the start of the coding region in one of the MMV007224-resistant clones [9,22]. Notably, among a small number of copy number variations (CNVs) identified in the compound **22**-resistant lines, clones isolated from flask 2 possessed a CNV of the phospholipid-translocating "flippase" *pfatp2* (PF3D7_1219600; Fig 4A and S6 Table). Consistent with this, clone 2.1 isolated from flask two had a modestly higher IC$_{50}$ for compound **22** compared to flask one (clone 1.1), despite sharing the same PfQRP1 R676I mutation (Fig 2B and 2C).

An association of PfATP2 with related quinoxaline-containing compounds was observed with previous resistance selection experiments with MMV007224. Three independent selections (R1-R3) with MMV007224 all yielded CNVs covering the *pfatp2* gene, with only R2 also containing an additional *pfqrp1* frameshift mutation at amino acid 100 out of 2126, resulting in a truncated protein [21]. To evaluate whether *pfatp2* amplification also confers resistance to compounds **22, 31** and **33**, we tested the MMV007224-resistant clones R1 and R2. Both lines conferred a ~2- to 3-fold increase in IC$_{50}$ for all three compounds, with compound **33** the least affected. This relatively small shift in potency was similar to the original selection compound MMV007224 (Fig 4B–4D). The modestly higher IC$_{50}$ values for line R2 that possesses both the *pfatp2* CNV and the *pfqrp1* frameshift mutation suggest that both mechanisms together may contribute to resistance to quinoxaline-based compounds.

## Loss-of-function mutations in PfQRP1 confer resistance

We next explored whether loss-of-function of the predicted hydrolase activity of PfQRP1 underlies resistance to these quinoxaline-based compounds. Residues corresponding to a potential catalytic triad – S1622/D1829/H2047 in *P. falciparum* – are highly conserved across Apicomplexan orthologs of PfQRP1 (Fig 5A) and are in close proximity in the AlphaFold model (Fig 5B). We generated a CRISPR-edited truncation mutant at the D1829 residue, inserting a stop codon. Truncation of PfQRP1 within the hydrolase domain resulted in an elevated IC$_{50}$ for compounds **22** and **31**, whereas compound **33** was not affected despite close structural similarity (Fig 5C and 5D). A more subtle D1829A point mutant, predicted to disrupt the putative catalytic triad, behaved similarly to the truncation mutant (Fig 5C and 5D). The related compounds MMV665794 and MMV007224 were also rendered less potent by both PfQRP1 mutations (Fig 5C and 5D). Collectively, these results suggest that loss of function in the PfQRP1 hydrolase domain is protective against this quinoxaline-based compound series.

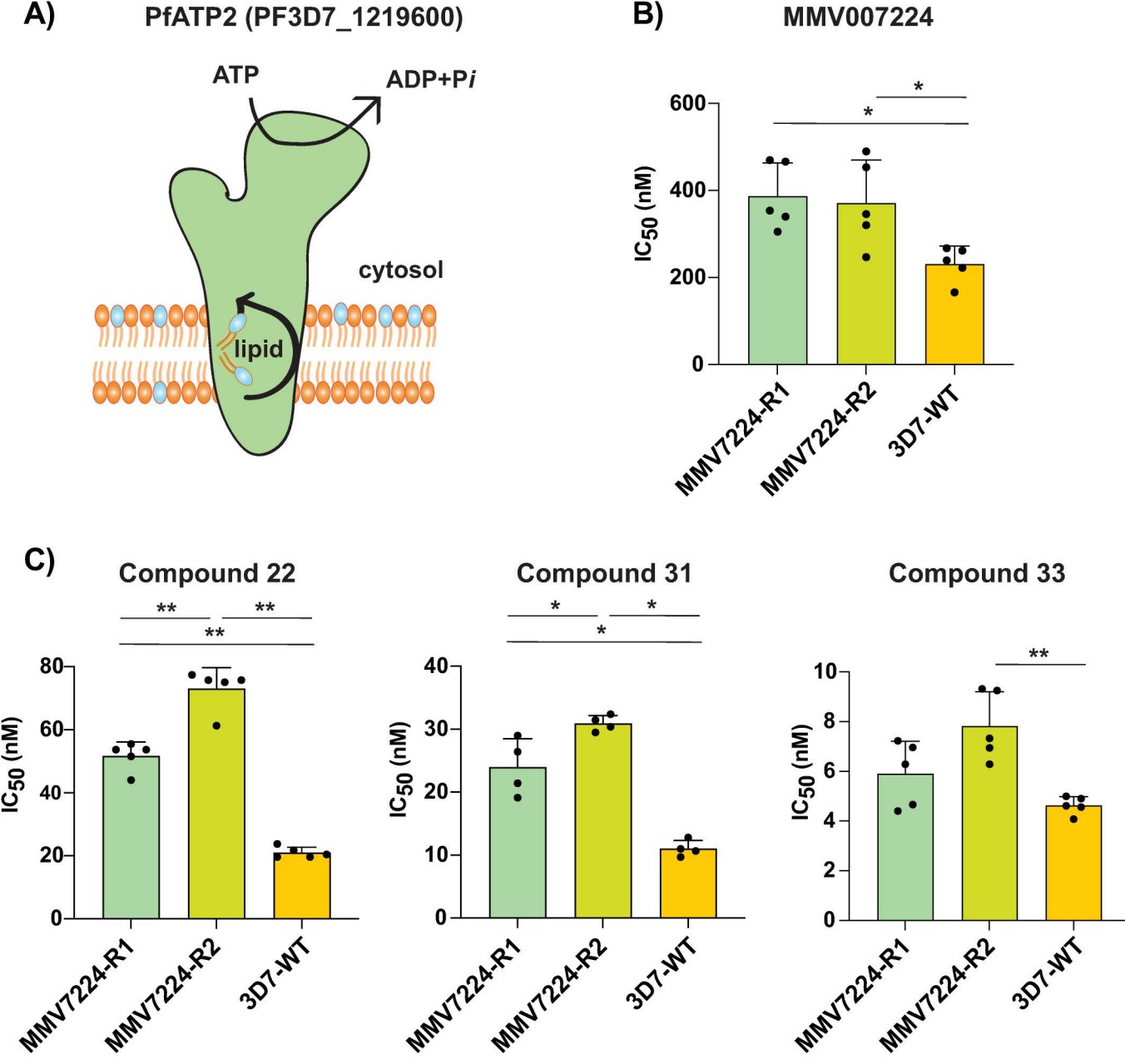

**Fig 4. Amplification of a phospholipid-translocating ATPase reduces susceptibility to quinoxaline-containing compounds.** (A) Model of PfATP2 lipid flippase. The P4-ATPase is predicted to contain 10 transmembrane segments, with phospholipid translocation from the luminal to the cytosolic leaflet of the membrane powered by ATP hydrolysis. (B–C) MMV007224-selected clones R1 (*pfatp2* CNV) and R2 (*pfatp2* CNV and *pfqrp1* frameshift) were tested against (B) MMV007224, and (C) compounds **22**, **31** and **33**. Each dot represents a biological replicate (n = 4–5) with mean±SD shown as bar charts, and statistical significance determined by Mann-Whitney *U* tests (*p < 0.05, **p < 0.01).

## Quinoxaline analogs interfere with hemoglobin breakdown but not hemozoin formation

Finally, as representatives of the quinoxaline series, we tested the ability of MMV007224 and MMV665794 to inhibit heme detoxification. We first evaluated the ability of these compounds to block formation of β-hematin (synthetic hemozoin) in a cell-free set-up that mimics the

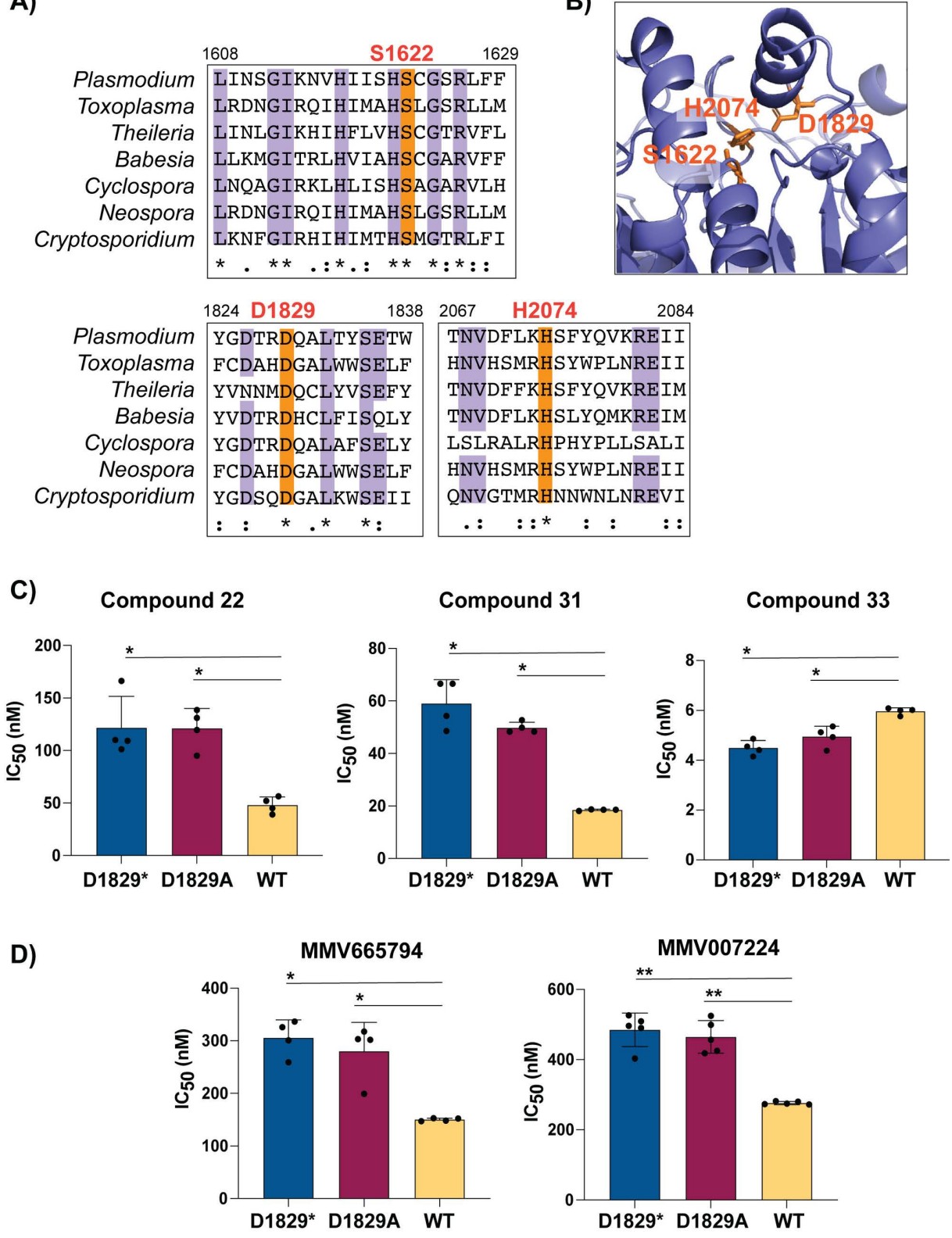

**Fig 5. Mutation of the putative catalytic triad of PfQRP1 confers resistance.** (**A**) Partial sequence alignment of *pfqrp1* homologs showing high conservation of putative catalytic triad residues (orange), comparing *P. falciparum pfqrp1* (PF3D7_1359900) with Apicomplexan orthologs listed in Fig 3. (**B**) AlphaFold model of PfQRP1 showing the S-D-H residues that form the putative catalytic triad. (**C**) CRISPR-edited

parasites with D1829 mutated to a stop codon (D1829*) or alanine (D1829A) show elevated $IC_{50}$ values for **(C)** compounds **22, 31** and **33**, and **(D)** MMV665794 and MMV007224. Each dot represents a biological replicate (n = 4–5) with mean±SD shown as bar charts, and statistical significance determined by Mann-Whitney *U* tests (*p < 0.05, **p < 0.01).

physiological micro-environment of the parasite digestive vacuole [23]. As expected, chloroquine potently inhibited β-hematin formation ($IC_{50}$ = 16 μM) (S7 Fig and S7 Table). In contrast, MMV007224 and MMV665794 had no effect on β-hematin formation in this system ($IC_{50}$ values > 150 μM), similar to the negative control KAE609.

We next asked whether the inhibition of heme detoxification by this series is only manifested intracellularly. To this end, we employed a cellular heme fractionation assay that quantifies the various species of heme fractionated as hemoglobin heme, 'free' heme and hemozoin heme from drug-treated cells [24]. This approach can discriminate "true inhibitors" that cause concentration-dependent increases in 'free' heme and corresponding decreases in hemozoin levels within the parasite digestive vacuole from non-inhibitors. MMV007224-treated parasites showed a modest concentration-dependent increase in undigested hemoglobin levels at treatment concentrations corresponding to 1× to 1.5× MMV007224 $IC_{50}$ (Fig 6A). A similar trend was observed for the MMV665794-treated parasites, peaking at concentrations equivalent to 1.5×$IC_{50}$ (Fig 6B). For both compounds, beyond their $IC_{50}$ concentration, hemoglobin levels then appeared to decline, possibly reflecting gradual parasite death and subsequent inability to import or degrade more hemoglobin. As further confirmation of the cell-free experiment results, there were no significant changes in the levels of 'free' heme or hemozoin relative to untreated controls. An overlay of 72 h parasite survival results with the 'free' heme levels showed no correlation, similar to the negative control KAE609 (Fig 6C and 6E), indicating that these quinoxaline analogs do not exert their anti-plasmodial activity by directly perturbing heme detoxification. In contrast, chloroquine-treated parasites exhibited significant concentration-dependent increases in 'free' heme that tracked with parasite inhibition and corresponded with decreases in hemozoin (Fig 6C and 6E).

## Discussion

Here, we report that a series of anti-schistosomal, quinoxaline-based compounds have potent activity against the asexual blood stage of *P. falciparum*. The compounds had sub-micromolar anti-plasmodial activity, reaching as low as single-digit nanomolar $IC_{50}$ values against standard laboratory lines 3D7 and Dd2, parasite strains from Uganda, Tanzania and Ghana, as well as multi-drug resistant Cambodian isolates. To understand the mode of action of this compound series, we performed *in vitro* resistance selections with three compounds (**22**, **31**, and **33**). We were only successful in generating resistant parasites against compound **22**, and not **31** or **33**, despite using a high inoculum of up to $10^9$ parasites of a mutator line with an elevated mutation rate, which we have shown to elicit resistance to previously resistance-refractory compounds [9]. The inability to generate resistance to compound **31** and **33** using these conditions, and to obtain only low-grade resistance to **22**, suggest a promising resistance profile for this class of compounds and its cognate target.

Whole-genome sequencing of compound **22**-evolved clones identified mutations in *pfqrp1*, a gene we recently characterised as encoding a quinoxaline-resistance protein [9]. Furthermore, the compound **22**-evolved clones showed cross-resistance with compounds **30**, **31**, **32** and **35** as well as related quinoxaline compounds MMV665794 and MMV007224. However, compound **33**, despite only differing from **31** by the positions of the trifluoromethyl groups and introduction of fluorine atoms at C5 of each *N*-aryl group, appeared insensitive to this

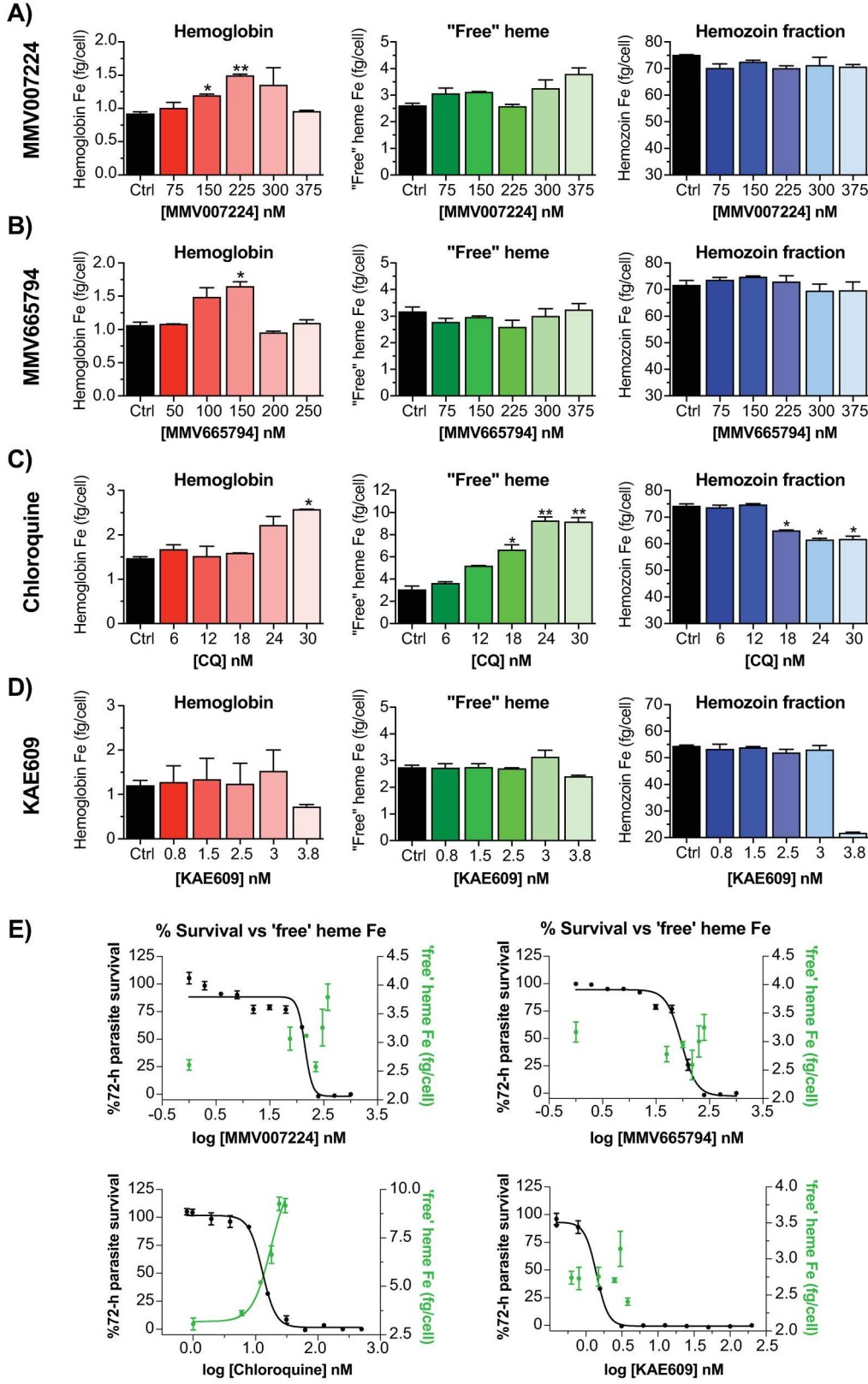

**Fig 6. Heme species fractioned from drug-treated parasites. (A–D)** Heme species are presented on the y-axis as undigested hemoglobin (*left*), 'free' heme (*middle*) and hemozoin (*right*) from treatment of 3D7 parasites with (A) MMV007224, **(B)** MMV665794, **(C)** chloroquine and **(D)** KAE609, respectively, at the various concentrations shown

in the x-axis. **(E)** Overlay of 72 h parasite survival with the total amount of 'free' heme Fe as a function of drug concentration. Data represent means of two biological repeats done in technical duplicates (N, **n** = 2, 2). Statistical comparisons were conducted by comparing different incubation concentrations with untreated controls using Students' t-test with Welch correction and significance set at *$p < 0.05$, **$p < 0.01$.

resistance mechanism. We next validated PfQRP1 as causal for resistance; CRISPR-edited lines bearing two resistance mutations to compound MMV665794 [9] showed cross-resistance to compounds **22** and **31**. PfQRP1 is a large protein of 250 kDa and is predicted to possess four N-terminal transmembrane segments as well as a putative C-terminal alpha/beta hydrolase domain. Although the mechanism of resistance is unknown, the compound **22**-resistance mutations, as well as the two previously identified mutations conferring resistance to MMV665794, map near to the putative hydrolase domain in the AlphaFold structure. In addition, the frameshift mutation identified in the MMV007224-resistant clone R2 suggests that resistance may be mediated by PfQRP1 loss-of-function.

To further explore whether the putative hydrolase function of PfQRP1 was related to its mechanism of resistance, we generated a point mutation in D1829, one of the putative catalytic triad residues. Conversion to alanine, or a more dramatic stop codon insertion, both conferred a 2–3 fold increase in IC$_{50}$ for compounds **22** and **31** (but not **33**), similar or greater in level of resistance to the drug-selected mutations, suggesting that loss of PfQRP1 hydrolase function mediates protection. Whether this is through direct action on the compounds, as with the PfPARE prodrug convertase [25], or a more indirect mechanism is unknown.

The non-essential nature of PfQRP1 suggests that the direct target of these quinoxaline-based compounds is likely another protein, with the phospholipid-transporting P4-ATPase, PfATP2 being a prime candidate. Copy number amplification of *pfatp2* was observed in a subset of compound **22**-selected clones, suggesting that increased abundance of this putative target is protective. Similarly, *pfatp2* CNVs were previously obtained with resistance selections using MMV007224 [21]. Our data showed that a MMV007224-resistant clone with a *pfatp2* CNV but no PfQRP1 mutation also confers cross-resistance to compounds **22, 31** and **33** described herein. The significance of the small number of other CNVs obtained in the drug-pressured lines is unknown, although the presence of an epigenetic regulator, *bdp4*, in one CNV might affect expression of genes related to the mode of action of these quinoxaline compounds.

In general, phospholipid-transporting "flippases" perform ATP-dependent translocation of phospholipids from the extracellular or luminal face of the lipid bilayer towards the cytoplasmic leaflet [26–28]. Unlike *pfqrp1*, *pfatp2* is likely an essential gene based on the absence of *piggyBac* insertions and the inability to disrupt the *P. berghei* ortholog [29,30]. Furthermore, a recent study by Qiu *et al.* localised PfATP2 to the parasite plasma membrane and demonstrated that conditional knockdown of *pfatp2* using the glmS ribozyme system arrested parasite growth [31]. Knockdown of PfATP2 levels caused sensitisation to MMV665794 and MMV007224, consistent with PfATP2 being their intracellular target. This study also showed that uptake of the fluorescent lipid substrate NBD-PS into saponin-lysed parasites was diminished both by knockdown of PfATP2, as well as by treatment with MMV665794 and MMV007224, suggesting that their mode of action is the perturbation of lipid internalisation.

In addition, lipid flippases typically require a CDC50 accessory subunit to function [26]. Of the three *cdc50* genes encoded by the *P. falciparum* genome, CDC50C was shown to be the sole essential ortholog for blood stage growth and the partner of PfATP2 by co-immunoprecipitation [32]. P-type ATPases more broadly are well-established drug targets, with Na$^+$-ATPase PfATP4 established as the target of cipargamin, which is in phase 2 clinical trials [33,34]. Whether *pfatp2* is essential at other stages of the lifecycle remains to

be determined; the blood-stage essentiality in *P. berghei* precluded further examination of mosquito and liver-stage phenotypes [30,35]. However, MMV007224 did not show activity against *P. berghei* liver stage parasites, suggesting that this compound series is not active in this phase of the parasite lifecycle [36]. More generally, perturbation of lipid dynamics may be an attractive and relatively unexplored area for antimalarial targets. A recent study identified the phospholipid transfer protein, PfSTART1, as the target of MMV006833, with inhibition impacting the expansion of the parasitophorous vacuole membrane [37].

The correlation between phenotypic activity against *Plasmodium* asexual blood stages and *Schistosoma* schistosomula, for the 11 compounds tested, is suggestive of a shared mode of action. This does not appear to be direct perturbation of heme detoxification. However, the observed influence of MMV007224 and MMV665794 on increased hemoglobin levels hints at potential disruption of optimal degradation of host hemoglobin, possibly as a consequence of altered membrane dynamics along the endocytic-vacuolar pathway. Evaluation of the 400 compounds from the Malaria Box against *S. mansoni* identified MMV007224 as among the most active hits against adult worms [38]. Notably, another of the most active compounds against adult *Schistosoma* from the Malaria Box was MMV665852, a N,N′-diarylurea compound. Resistance selections in *P. falciparum* with MMV665852 also yielded CNVs in *pfatp2* [21]. Although we could not identify a clear homolog of PfQRP1 in *S. mansoni*, there are multiple phospholipid-transporting ATPases annotated in this genome. InterPro domain IDs (IPR001757 and IPR006539 for a generic P-type ATPase and type 4 ATPase, respectively) identify 20 P-type ATPases in *S. mansoni*, of which 6 belong to the P4-ATPase subfamily that specialise in lipid rather than ion transport (S8 Fig). These may warrant further investigation as a new target class.

In summary, we describe the dual-parasite activity of a quinoxaline-core compound series that is potent against both *P. falciparum* and *S. mansoni*. *In vitro* evolution of resistance using a *P. falciparum* mutator line indicates this series has a low propensity for resistance, and when achieved, only a modest loss of potency. Resistance can be driven by mutations in a quinoxaline-resistance protein, PfQRP1, or CNVs of a phospholipid-transporting ATPase, PfATP2. The correlation in phenotypic activity between *Plasmodium* and *Schistosoma* hint at a shared mode of action and potential new targets for controlling the parasites responsible for two important infectious diseases.

## Materials and methods

### Ethics statement

Parasites were cultured in fresh human erythrocytes obtained with ethical approval from anonymous healthy donors, with informed written consent as part of the recruitment process, from the National Health Services Blood and Transplant (NHSBT) or the Scottish National Blood Transfusion Service (SNBTS). The use of erythrocytes was with approval from NHS Cambridgeshire Research Ethics Committee (15/EE/0253) and the Wellcome Sanger Institute Human Materials and Data Management Committee, and the University of Dundee Schools of Medicine and Life Sciences Research Ethics Committee (21/39).

### Parasite cultivation and transfection

*P. falciparum* parasites were cultured at 3% hematocrit in RPMI 1640 (Gibco) medium consisting of 0.5% Albumax II (Gibco), 25 mM HEPES (Sigma), 0.21% sodium bicarbonate, 1x GlutaMAX (Gibco), 25 µg/mL gentamicin (Gibco). A tri-gas mixture (1% $O_2$, 3% $CO_2$, and 96% $N_2$) was used for parasite cultures. Parasites were synchronized using a sorbitol and Percoll density gradient method [39]. The following parasite strains were used: 3D7, Dd2

(MR4, MRA-156, T. Wellems), GB4 (MR4, MRA-925, T. Wellems), UG659 (P. Rosenthal [40]), Tanzania (MR4, MRA-1169, Michal Fried), Cam/PH0212C (R. Fairhurst [41]), and Cam3.II/ PH0306-C (R. Fairhurst [42]). Transfection was performed using ring-stage parasites (5–8% parasitemia) using a Gene Pulser Xcell (Biorad) electroporator.

To generate the PfQRP1 hydrolase mutants (D1829stop and D1829A), the pDC2-Cas9-gRNA-donor plasmid [43] was transfected into the *P. falciparum* Dd2 line. Donor plasmid (50 µg) containing the sgRNA (AGCTTTAACATATTCAGAAA) and a *pfqrp1* homology region of 672 bp, centred on the D1829 residue, was used for transfection. Transfectants were selected using 5 nM WR99210 (a potent inhibitor of *P. falciparum* dihydrofolate reductase (DHFR) provided by Jacobus Pharmaceuticals) for 8 days, followed by culturing without drug until parasites were observed. Parasite clones were obtained using limiting dilution cloning. Confirmation of gene editing was performed with allele-specific PCR (forward primer CTGAAGAAGATGAATGGGAACA and reverse primer CCACCTTCTCCTTCACCAAC) and Sanger sequencing using an internal primer (TGGAAGAAAGGAAAACACAACA).

### *In vitro* drug resistance selections using Dd2-Polδ

*In vitro* evolution of resistance was performed against compounds **22**, **31** and **33**. Resistance selections were performed using a mutator parasite line Dd2-Polδ with an elevated mutation rate [9]. Three independent flasks with either $1 \times 10^8$ or $1 \times 10^9$ parasites each were treated with $5 \times IC_{50}$, and parasite death was monitored by microscopy (Giemsa staining). Viable parasites were undetectable by blood smear after eight days of compound treatment. After eight days, compounds were removed from the media and the media was changed on alternate days. Parasites reappeared after approximately 3 weeks from the washout of the drug (Fig 2A). For recrudescent cultures, compound susceptibility was determined by dose-response assays. Parasite clones were obtained by limiting dilution and harvested for genomic DNA extraction, followed by whole-genome sequencing.

### Minimum inoculum for resistance (MIR)

For minimum inoculum for resistance (MIR) assays, three independent flasks of either Dd2 parental parasites or the isogenic Dd2-Polδ mutator line were exposed to $3 \times IC_{50}$ of either compound **22** (90 nM) or a positive control compound GNF179 (10.5 nM) [36]. The parasite inoculum for each culture ranged from $10^4$, $10^5$, $10^6$, $10^7$ to $10^8$ parasites. For compound **22**, the concentration was raised to $4 \times IC_{50}$ (120 nM) at day 6 and finally $5 \times IC_{50}$ (150 nM) at day 9 (see S6 Fig) in order to fully clear the culture. Selections were stopped after 45 days (GNF179) or 54 days (compound **22**).

### Stage specificity assay

Stage-specific drug assays were performed following a protocol modified from Murithi et al. [44]. In this assay, parasites were tightly synchronized to the ring stage with repeated 5% sorbitol treatments. On the assay day, schizonts were purified using a 63% Percoll gradient and allowed to reinvade. After 6 h of incubation, the culture was synchronized again with 5% sorbitol to obtain pure rings. Parasites were then plated in 96-well plates and exposed to different drug concentrations at various stages: Ring (6–16 h), Trophozoite (18–28 h), and Schizont (30–40 h). Drug treatment was removed through three rounds of washes. Plates were lysed 62 h after drug washout, and parasite growth was assessed similarly to the standard 72 h assay that used constant drug exposure.

## Compound dose-response assay

Compound dose-response assays were performed in flat-bottom 96-well plates. Tightly synchronized ring-stage parasites were diluted to 1% parasitemia at 2% hematocrit, and incubated with a two-fold serial dilution of compounds in complete medium. Untreated parasites or uninfected RBCs were included in the assay plate as controls. Parasite growth was assessed after 72 h by lysing parasites using 2× lysis buffer consisting of 10 mM Tris-HCl, 5 mM EDTA, 0.1% w/v saponin, and 1% v/v Triton X-100, supplemented with 2× SYBR Green I (Molecular Probes). The fluorescence was measured using a FluorStar Omega v5.11 plate reader. $IC_{50}$ analysis was performed using GraphPad Prism v9, and statistical significance was determined by a two-sided Mann-Whitney *U* test. All assays were performed in technical duplicate with at least three biological replicates, as noted in the figure legends.

## Rate of killing

The rate of killing was evaluated as previously described in Linares *et al.* [14]. Briefly, unlabelled RBCs infected with the Dd2 strain were treated with the compound at $10× IC_{50}$ for 24 and 48 h, with the compound refreshed after the first 24 h. After treatment, the compound was removed, and the culture was diluted (1/3) using fresh RBCs (2% haematocrit) pre-labelled with CFDA-SE (carboxylfluorescein diacetate succinimidyl ester, Life Technologies). Following a 48 h incubation under standard conditions, the ability of treated parasites to infect fresh labelled RBCs was measured using a Cytoflex flow cytometer by quantifying double-positive RBCs after labelling the parasites with MitoTracker Deep Red. Parasite viability was determined as the percentage of infected CFDA-SE-labelled RBCs in drug-treated samples at 24 and 48 h, compared to equivalent untreated controls. Standard compounds dihydroartemisinin, chloroquine, and atovaquone were used as benchmarks for comparing the killing rate of the test compounds.

## Prediction of ADME properties

Rule of five (Ro5) analysis was performed using SwissADME (available on http://www.swissadme.ch; [17]) to calculate each compound's molecular weight (MW), calculated Log P (cLogP), number of hydrogen bond donors (HBD), number of hydrogen bond acceptors (HBA) and number of rotatable bonds (NROTB). Predicted ADME parameters were calculated using QikProp 5.8 tool (Schrodinger 2018, New York).

## Detergent-based β-hematin inhibition assay

Drug stocks, including the control compounds chloroquine and KAE609, were reconstituted to 10 mM. From this stock 40 μL of each was delivered to wells in the final column (column 12) of a 96-well plate together with distilled water (120 μL) and NP40 detergent (305.5 mM, 40 μL). A solution containing water/NP40 (305.5 mM)/DMSO at a v/v ratio of 70%/20%/10% respectively was prepared and 100 μL was added to all other wells (columns 1–11). A serial dilution of each compound (100 μL) from column 12 to column 2 was carried out. Column 1 served as a blank with 0 mM sample. A 25 mM hematin stock solution was prepared by sonicating hemin in DMSO for one minute and then suspending 179 μL of this in a 1 M acetate buffer (20 mL, pH 4.8). The homogenous suspension (100 μL) was then added to the wells to give final buffer and hematin concentrations of 0.5 M and 100 mM respectively. The plate was covered and incubated at 37°C for 5 h. A solution of 50% (v/v) pyridine, 30% (v/v) $H_2O$, 20% (v/v) acetone and 2 M HEPES buffer (pH 7.4) was prepared, and 32 μL added to each well to give a final pyridine concentration of 5% (v/v). Acetone (60 μL) was then added to assist with hematin dispersion. The UV-vis absorbance of the plate wells was read on a SpectraMax P340

plate reader. Sigmoidal dose-response curves were fitted to the absorbance data using Graph-Pad Prism version 8 to obtain a 50% inhibitory concentration ($IC_{50}$) for each compound.

## Cellular heme fractionation assay

The baseline *in vitro* sensitivity of 3D7 to MMV007224, MMV665794, chloroquine and KAE609 was determined using a standard 72 h SYBR Green chemosensitivity assay. These results were used to determine the $IC_{50}$ levels that would inform the concentrations used to incubate the parasites. Cellular heme fractionation to measure the amounts of different heme species was thereafter conducted as previously described [24]. Briefly, ring-stage 3D7 parasites were treated for two cycles using 5% sorbitol to obtain synchronized parasites ~3–5 h post-invasion. Using the $IC_{50}$ determined in the 72 h chemosensitivity assay, parasites were incubated in a gradient of $IC_{50}$ concentrations ranging from 0.5 – 2.5× at 5% parasitemia and 2% hematocrit, with a no-drug control included. After 30 h, late trophozoites/early schizonts were harvested by lysis of the RBCs with 0.05% saponin followed by multiple washes with 1× PBS (pH 7.5) to remove traces of the RBC hemoglobin. Pellets were then resuspended in 1× PBS (pH 7.5) and an aliquot of the trophozoite suspension used to quantify the total number of trophozoites isolated using flow cytometry. Contents of the remaining trophozoite pellet were then released by hypotonic lysis and sonication. Following centrifugation, treatment with HEPES buffer (pH 7.4), 4% SDS, 25% pyridine solution, and 0.3 M NaOH, the fractions corresponding to undigested hemoglobin, "free" heme and hemozoin were carefully recovered. The UV–visible spectrum of each heme fraction as an Fe(III)heme–pyridine complex was measured using a multi-well plate reader (Spectramax 340PC; Molecular Devices). The total amount of each heme species was quantified using a heme standard curve whereby the mass of each heme-Fe species per trophozoite (fg/cell) was calculated by dividing the total amount of each heme species by the corresponding number of parasites in that fraction as determined by flow cytometry. Statistical comparisons and analyses for trends were made on GraphPad Prism version 10 using Students' t-test with Welch correction.

## Heme standard curve determination

The total amount of heme obtained from each fraction was quantified using a standard curve prepared from a 100 µg/mL heme standard solution of hematin (porcine) in 0.3 M NaOH. Serial dilutions of this standard were carried out in a 96-well plate with 100 µL 0.3 M NaOH as a blank. Fifty microlitres (50 µL) of each of the following solutions was added to 100 µL of hematin standard: 0.2 M HEPES pH 7.5, 4% (w/v) SDS, 0.3 M NaCl, 0.3 M HCl, 25% pyridine in 0.2 M HEPES pH 7.5 and water. The visible spectra of heme as Fe(III)heme-pyridine complex were recorded in a multi-well plate reader. The amount of heme Fe per cell was calculated by dividing the total heme Fe in each fraction by the number of cells determined in each fraction.

## Whole-genome sequencing

Parasites were harvested using 0.1% saponin lysis buffer of RBCs, followed by three washes of PBS (pH 7.5). Genomic DNA was extracted using the DNAeasy Blood and Tissue Kit (Qiagen). Genomic DNA concentration was quantified using a Qubit dsDNA BR assay kit and measured using a Qubit 2.0 fluorometer (Thermo Fisher Scientific).

## Single nucleotide variant and copy number variant calling

Whole-genome sequencing was performed using an IDT-ILMN Nextera DNA UD library kit and multiplexed on a NextSeq flow cell to generate 150 bp paired-end reads. Sequences

were aligned to the *Pf* 3D7 reference genome (PlasmoDB-48; https://plasmodb.org/plasmo/app/downloads/release-48/Pfalciparum3D7/fasta/) using the Burrow-Wheeler Alignment (BWA version 0.7.17). PCR duplicates and unmapped reads were filtered out using Samtools (version 1.13) and Picard MarkDuplicates (GATK version 4.2.2). Base quality scores were recalibrated using GATK BaseRecalibrator (GATK version 4.2.2). GATK HaplotypeCaller (GATK version 4.2.2) was used to identify all possible single nucleotide variants (SNVs), filtered based on quality scores (variant quality as function of depth QD > 1.5, mapping quality > 40, min base quality score > 18, read depth > 5) and annotated using SnpEff version 4.3t [45]. Comparative SNP analyses between eight drug-treated Dd2-Polδ clones and the Dd2-Polδ parental strain were performed to generate the final list of SNPs (S5 Table). BIC-Seq version 1.1.2 [46] was used to discover copy number variants (CNVs; S6 Table) against the Dd2-Polδ parental strain using the Bayesian statistical model. SNPs and CNVs were visually inspected and verified using Integrative Genome Viewer (IGV). All gene annotations in the analysis were based on PlasmoDB-48 (https://plasmodb.org/plasmo/app/downloads/release-48/Pfalciparum3D7/gff/).

## Supporting information

**S1 Fig. Structures of compounds.**
(PDF)

**S2 Fig. Stage-specificity profiling.** Stage-specific dose response assays for compounds **(A) 22**, **(B) 31**, and **(C) 33**. Assays were performed on tightly synchronised cultures, with compound applied for 10 h at the ring (6–16 h post-invasion), trophozoite (18–28 h post-invasion) and schizont (30–40 h post-invasion) stages. After drug washout, cultures were incubated for a further 62 h. For comparison, assays were also performed with constant drug pressure applied for the full 72 h. The bar graph (*left*) represents mean $IC_{50}$ values for three replicates, the survival graphs (*right*) display the dose response curves from three independent experiments. Error bars indicate the standard error of the mean based on these three independent repeats.
(PDF)

**S3 Fig. Parasite killing rate.** Rate of parasite killing for compounds **22**, **31** and **33** was examined by treating Dd2 parasites in unlabelled RBCs with $10 \times IC_{50}$ for 24 and 48 h, followed by washout. Reinvasion of viable parasites into fresh RBCs pre-labelled with carboxylfluorescein diacetate succinimidyl ester (CFDA-SE) was measured after a further 48 h, and the percentage of remaining viable parasites compared with untreated controls. For comparison, standard antimalarials included fast-killing dihydroartemisinin, chloroquine and slow-killing atovaquone. The average (±SD) of three biological replicates (each with two technical replicates) is shown.
(PDF)

**S4 Fig. Selectivity of compounds. (A)** Comparison of compound potency between (*left*) the larval stage of *S. mansoni* and *P. falciparum* 3D7 strain, and (*right*) HepG2 and *P. falciparum* 3D7. Data for *S. mansoni* and HepG2 were derived from [13] and are shown in S2 Table. Correlation was performed using Spearman's correlation test. **(B)** Selectivity index of compounds on *P. falciparum* 3D7 or Dd2 strains was calculated relative to HepG2 72 h $CC_{50}$ values (S2 Table).
(PDF)

**S5 Fig. Cross-resistance of compound 22-evolved clones.** Clones derived from *in vitro* resistance evolution with compound **22** (see Fig 2) were evaluated against additional compounds from this series, compounds **30**, **32**, **33** and **35**. Compound **30** is a structural isomer of compound **31** (Fig 2b) and compounds **32** and **33** are isomers of each other. Each dot represents

a biological replicate (n = 5) with mean±SD shown as bar chart, and statistical significance determined by Mann-Whitney *U* test (*p < 0.05, **p < 0.01).
(PDF)

**S6 Fig. Minimum inoculum for resistance (MIR).** Three independent cultures of either the parental Dd2 or isogenic Dd2-Polδ mutator lines were exposed to compound **22** or GNF179. Parasite inoculum ranged from $10^4$ to $10^8$ parasites per culture. Drug pressure was initiated at $3\times IC_{50}$ for both compound **22** (90 nM) and GNF179 (10.5 nM). For compound **22**, the selection pressure was raised to $4\times IC_{50}$ (120 nM) at day 6 and $5\times IC_{50}$ (150 nM) at day 9 in order to fully clear the culture. Red squares indicate absence of parasites after 45 days (GNF179) or 54 days (compound **22**), green squares indicate recrudescence, with the day of recrudescence representing the day when parasitemia crossed 1%.
(PDF)

**S7 Fig. β-hematin inhibition assay for quinoxaline analogs and antimalarial control compounds.** Representative plot showing inhibition of β-hematin formation versus concentration profiles for MMV007224 (black), MMV665794 (red), chloroquine (blue) and KAE609 (green). Adjacent $IC_{50}$ data represent mean values of two biological replicates performed with technical duplicates. Data were fitted to the sigmoidal concentration response (variable slope) equation in GraphPad Prism to determine the $IC_{50}$.
(PDF)

**S8 Fig. Sequence alignment of phospholipid flippases.** (**A**) Lipid flippases used in the alignment. (**B**) Sequence alignment of human ATP8A2, *P. falciparum* PfATP2, and six putative lipid-translocating ATPases from *S. mansoni*. The actuator (A), nucleotide binding (N), and phosphorylation (P) domains are shown, as well as the first six transmembrane segments (M1-6). Key conserved residues D (in A domain) and D (P domain) involved in the phosphorylation (DKTGT) and dephosphorylation (DGET) cycle are highlighted by a star. The purple circles highlight the conserved N and I residues located in M4 domain that are important for recognition and release of lipid, respectively. The green triangle indicates the K residues in the M5 domain required for the sensitivity to the lipid subtype [26,47].
(PDF)

**S1 Table. Compound information.**
(XLSX)

**S2 Table. Activity against *Plasmodium* asexual blood stages, *Schistosoma* schistosomula and HepG2 cells.**
(XLSX)

**S3 Table. Lipinski rule of five criteria for compound series.**
(XLSX)

**S4 Table. Prediction of ADME properties.**
(XLSX)

**S5 Table. Single-nucleotide variants for compound 22-selected clones.**
(XLSX)

**S6 Table. Copy number variants from compound 22-selected clones.**
(XLSX)

**S7 Table. ß-hematin inhibition assay.**
(XLSX)

## Acknowledgments

We would like to thank R. Fairhurst for providing the Cambodian strains. We are grateful to members of the Lee lab and the reviewers for constructive feedback.

## Author contributions

**Conceptualization:** Mukul Rawat, Gilda Padalino, Karl F. Hoffmann, Marcus C. S. Lee.

**Formal analysis:** Tomas Yeo.

**Funding acquisition:** David A. Fidock, Karl F. Hoffmann, Marcus C. S. Lee.

**Investigation:** Mukul Rawat, Edem Adika, John Okombo, Tomas Yeo, Marcus C. S. Lee.

**Resources:** David A. Fidock, Karl F. Hoffmann.

**Supervision:** Andrea Brancale, David A. Fidock, Karl F. Hoffmann, Marcus C. S. Lee.

**Writing – original draft:** Mukul Rawat, Gilda Padalino, David A. Fidock, Karl F. Hoffmann, Marcus C. S. Lee.

**Writing – review & editing:** Mukul Rawat, Gilda Padalino, Edem Adika, John Okombo, Tomas Yeo, Andrea Brancale, David A. Fidock, Karl F. Hoffmann, Marcus C. S. Lee.

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
