## [Decision Letter · Decision Letter 0]

4 Jun 2024

Dear Professor Lee

Thank you very much for submitting your manuscript: Quinoxaline-Based Anti-Schistosomal Compounds Have Potent Anti-Malarial Activity: PPathogens-D-24-00857 for review by PloS Pathogens. Your manuscript was fully evaluated at the editorial level and by independent peer reviewers. The reviewers appreciated the attention to an important problem, but raised substantial concerns about the manuscript as it currently stands.

- Referee 1 points out that it is preferable to first test the ease of selection of resistance using a standard approach, with lower concentrations of compound against a varied number of parasites. This would allow a comparison using a standard approach and using atovaquone as control.

- Referee 3 points out that the identification of PfATP2 as the likely target of the quinoxalines is quite weak because it relies on the demonstration of PfATP2 being essential which is not conclusive. This could be due to technical errors with transfection experiments and targeting vector for the P. berghei homologue of PF3D7_1219600 which is annotated with short recombination arms that could reduce recombination at the locus.

- Referee 2 also points out that Compound 33 was not tested against gene-editing parasites: this is important because the Cambodian strains appear to show differences in their profiles for 21 and 31. Note the typo with PfQRP2 in this critique that should be PfQPR1.

Overall these are significant concerns that relate to the description and identification of the mechanisms of resistance that should be addressed in a Major Revision. These issues must be addressed before we consider a revised version of your study. We cannot, of course, promise publication at that time. We ask you to modify the manuscript according to the review recommendations before we can consider your manuscript for acceptance.

When you are ready to resubmit, please be prepared to provide the following:

(1) A letter containing a detailed list of your responses to the review comments and a description of the changes you have made in the manuscript.

(2) Two versions of the manuscript: one with either highlights or tracked changes denoting where the ext has been changed; the other a clean version (uploaded as the manuscript file).

We hope to receive your revised manuscript within 60 days. If you anticipate any delay in its return, we ask that you let us know the expected resubmission date by replying to this email. Revised manuscripts received beyond 60 days may require evaluation and peer review similar to that applied to newly submitted manuscripts.

We cannot make any decision about publication until we have seen the revised manuscript and your response to the reviewers' comments. Your revised manuscript is also likely to be sent to reviewers for further evaluation.

Sincerely,

Richard J. Martin, BVSc, PhD, DSc, DipECVPT, FRCVS

Guest Editor

PLOS Pathogens

Dominique Soldati-Favre

Section Editor

PLOS Pathogens

Michael Malim

Editor-in-Chief

PLOS Pathogens

orcid.org/0000-0002-7699-2064

PPathogens-D-24-00857

Quinoxaline-Based Anti-Schistosomal Compounds Have Potent Anti-Malarial Activity

Reviewer's Responses to Questions

**Part I - Summary**

Reviewer #1: This report addresses antimalarial properties of quinoxalines originally developed to treat schistosomiasis. The MS is well-written, clearly describes potent compounds with limited ability to select for parasite resistance, and offers a partial explanation of the mechanism of resistance to one compound. The conclusions build on prior work by the authors in which they studied selection of resistance by related compounds. Some concerns are as follows.

Reviewer #2: This manuscript by Rawat et al describes the discovery of potent antiplasmodial compounds based on quinoxaline-based schistosomal inhibitors. In fact, two of the analogues, 31 and 33, exhibit impressive single digit nM potency against both Dd2 and 3D7 as well as artemisinin partial resistant Cambodian strains. These compounds were shown to be irresistible even in the mutator strain. While the compound 22 which is slightly less potent generated modest resistance with point mutations in a non-essential gene PfQRP1. Although loss-of-function mutations suggest a mechanism of action via PfQRP1, compound 22 also yielded CNVs PfATP2 implying its role. In summary, the work presented describes the identification of potent antiplasmodial compounds with high barrier to resistance exhibiting complex yet novel mechanism of action that is distinct from that of current antimalarials.

Strengths: The research presented here is an important contribution to the malaria drug discovery efforts. Focus of the work is clearly to define the mechanism of action of these promising antiplasmodials.

Weakness: In vivo efficacy of these promising compounds would have made the story complete.

Reviewer #3: The study identifies a novel chemotype - quinoxaline as having anti-malarial activity. The manuscript describes the anti-Plasmodial activity, a mechanism of resistance and a putative mechanism of action of quinoxaline-containing compounds. Quinoxaline-containing compounds are contained in the MMV Malaria Box, suggesting this chemotype has potential for anti-malaria drug development. The quinoxaline-containing compounds interrogated in this study were originally identified for their anti-Schistosomal activity. Hence this study is an interesting example of pathogen hopping. The authors tested their collection of quinoxaline-containing anit-Schisto compounds and found that 3 are active against the P. falciparum lab strain, 3D7. One of the 3 has limited activity against drug-resistant strains, the second maintains moderate activity and the third is highly active. In an attempt to gain insight into potential modes of action and mechanisms of resistance, the authors conducted evolved resistance studies using the 3 compounds. Resistance could not be selected against the most potent compound. Using parasites resistant to the other 2 compounds, the authors propose PfQRP1 as a mediator of resistance and PFATP2 as the direct target of the compounds. However there is no activity assays with PFATP2 to support its role as a target of this novel chemotype.

**Part II – Major Issues: Key Experiments Required for Acceptance**

Reviewer #1: 1) The antimalarial properties of the compounds studied were confined to simple in vitro IC50 determinations. In depth consideration of antimalarial or drug-like properties of the compounds were not considered. An antimalarial drug discovery report would typically contain a good deal more data on activity, e.g. activity against additional parasite strains, stage specificity of activity, rate of killing, activity against other parasite stages, and/or in vivo activity in a murine model. Enthusiasm for this compound series (and thus interest in studies of mechanisms of resistance) would be greater if the promise of the class as potential antimalarials was better established.

2) Similarities between inhibition of malaria parasites and schistosome worms is interesting, but this begs questions about the specificity of the inhibitors. Malaria parasites are arguably as different from schistosomes as both organisms are from humans. The manuscript offers no information, either from described new experiments or the literature, on the likelihood that the studied quinoxalines will exert toxicity in humans, as they do in protozoans and worms. Indeed, this omission is glaring and almost seems intentional; one suspects that there are skeletons in the closet. Some description of data suggesting a therapeutic window, such that quinoxalines can be seriously considered as drugs, would be very helpful. No toxicity data or other considerations of drug-like (ADME) properties of the compounds is included in this report, but if some such data was included in prior studies of schistosomes (or any other literature), reference to that literature would be helpful.

3) Consideration of ease of resistance selection and insights into resistance mechanisms form the bulk of the manuscript. It is not clear why the authors did not study selection of resistance in standard P. falciparum strains, but rather only in an engineered strain. This strain is a great tool to study compounds for which resistance cannot easily be selected (as appears to be the case with the quinoxalines), but it would have been preferable to first test ease of selection of resistance using a standard approach, with a somewhat lower concentration of compound tested against varied numbers of parasites; excellent, now standardized approaches for such analyses have been developed by one of the authors (David Fidock). Although the results certainly suggest that the tested compounds do not easily select for resistance, it would have been helpful to have direct comparison of ease of selection using a standard approach, which would have allowed comparison with a large literature describing similar experiments with dozens of other compounds, and also to include controls (e.g. atovaquone, which selects easily) in the described experiments. Nonetheless, the selection experiments were of value, as successful selection of resistance with compound 22 enabled studies of mechanisms of resistance selection.

Reviewer #2: None.

Reviewer #3: The manuscript is clearly written and data presentation is excellent.

My major concern centers on the identification of PfATP2 as the likely target. While additional experimentation may show this to be true, the current case is quite weak. It relies of the putative essentiality of PfATP2, which is itself is less than conclusive. Negative results in transfection experiments can be due to technical issues rather than essential function of the target gene. In deed the targeting vector for P. berghei homolog of PF3D7_1219600 is annotated as having short recombination arms that can easily result in a lower rate of recombination at the locus. Similarly, piggybac transposon mutagenesis in P. falciparum has many false positives. The authors can conduct robust genetic targeting of PFATP2 (TetR=DOZI, DiCre) to determine whether or not it is essential. In addition to demonstrating essentiality of PFATP2, functional assays are needed to demonstrate that it is the target of the compounds, at least in vitro.

Another concern is the extent to which PfQRP2 is responsible for resistance to compounds 22, 31 and 33. Compound 33 was not tested against gene-edited parasites. 22 and 31 have quite different profiles in the Cambodian strains suggesting multiple mechanisms of resistance, some shared and others unique to 31. What is the status of PfQRP2 and PfATP2 in the Cambodian strains? If PfQRP2 mutations are absent in these strains then the significance of PfQRP2 as a resistance-mediator is reduced.

**Part III – Minor Issues: Editorial and Data Presentation Modifications**

Reviewer #1: 1) Line 26. “Evolution of resistance” is shorthand that will not be understood by many readers. Wording should be changed to explain that experiments involved attempts to select for resistance in vitro.

2) Line 43. “New combinations of drugs” is not a strategy. The strategy is to utilize new drugs for the treatment or prevention of malaria.

3) Line 45. The development of effective drugs does not require identification of targets. Many drugs that effectively treat malaria and other infections do not have known targets.

4) Line 53. As noted above, “in vitro evolution of resistance” should be explained with first usage in the text.

5) Figure 1. In the chemical structures shown, the orientation of compound 31 is flipped compared to that of the other structures; this won’t bother synthetic chemists, but makes comparison a bit more difficult for non-chemists. Showing the structures in the same orientation is preferred.

6) Line 147. “Only one gene, PF3D7_1359900, was common to all clones” is obviously false; all clones shared thousands of genes. Rather, the authors meant to say that only one gene had unique mutations in all resistant clones.

Reviewer #2: The compounds should be referred to as antiplasmodials (including the title) as no antimalarial property has been established in the rodent model.

Reviewer #3: (No Response)

PLOS authors have the option to publish the peer review history of their article (what does this mean? ). If published, this will include your full peer review and any attached files.

**Do you want your identity to be public for this peer review?** For information about this choice, including consent withdrawal, please see our Privacy Policy .

Reviewer #1: No

Reviewer #2: No

Reviewer #3: No
---

## [Decision Letter · Decision Letter 1]

8 Dec 2024

PPATHOGENS-D-24-00857R1Quinoxaline-Based Anti-Schistosomal Compounds Have Potent Anti-Plasmodial ActivityPLOS Pathogens Dear Dr. Lee, Thank you for submitting your manuscript to PLOS Pathogens. After careful consideration, we feel that it has merit but does not fully meet PLOS Pathogens's publication criteria as it currently stands. Therefore, we invite you to submit a revised version of the manuscript that addresses the points raised during the review process. Please submit your revised manuscript within 60 days Feb 06 2025 11:59PM. If you will need more time than this to complete your revisions, please reply to this message or contact the journal office at plospathogens@plos.org. Please include the following items when submitting your revised manuscript: * A rebuttal letter that responds to each point raised by the editor and reviewer(s). You should upload this letter as a separate file labeled 'Response to Reviewers '. This file does not need to include responses to any formatting updates and technical items listed in the 'Journal Requirements' section below.* A marked-up copy of your manuscript that highlights changes made to the original version. You should upload this as a separate file labeled 'Revised Manuscript with Track Changes '.* An unmarked version of your revised paper without tracked changes. You should upload this as a separate file labeled 'Manuscript '. If you would like to make changes to your financial disclosure, competing interests statement, or data availability statement, please make these updates within the submission form at the time of resubmission. Guidelines for resubmitting your figure files are available below the reviewer comments at the end of this letter. We look forward to receiving your revised manuscript. Kind regards,Richard J. Martin, BVSc, PhD, DSc, DipECVPT, FRCVSGuest EditorPLOS Pathogens Dominique Soldati-FavreSection EditorPLOS Pathogens Michael Malim

Editor-in-Chief

PLOS Pathogens

orcid.org/0000-0002-7699-2064 **Additional Editor Comments:** The MS has been reviewed again following revision and resubmission with the author’s response to reviewers.A concern of original reviewer 1 was that the mechanisms of resistance identified were incomplete: Studies of likelihood of resistance and mechanisms of resistance are of important for compounds likely to be advanced as antimalarial drug candidates. The reviewer had concerns that that the mechanisms of resistance of the studied quinoxalines were not sufficient.A similar concern was expressed by original reviewer 3 and expressed again following revision 1. The concerns were whether PfATP2 is essential and the actual target of these compounds. Also, there were concerns that there was no biochemical assay, and the genetic data cited was weak because the essential nature of CDPK1 not being confirmed.I have looked again to search for recent bioRxiv for Adele Lehane and PfATP2 but not found it to support the evidence that PfATP2 is essential. Thus, it is my opinion that the manuscript as it stands, without clearer evidence of the mechanisms of resistance, does not fully meet PLoS Pathogen’s publication criteria as it stands.1) The authors should genetically target PFATP2 (e.g. TetR=DOZI, DiCre) to determine if it is essential as has been suggested by reviewer 3 and alluded to by the original referee 1.2) The authors should also include a relevant biochemical assay showing that it is the target of the compounds. **Journal Requirements:**

1) We ask that a manuscript source file is provided at Revision. Please upload your manuscript file as a .doc, .docx, .rtf or .tex. If you are providing a .tex file, please upload it under the item type LaTeX Source File and leave your .pdf version as the item type Manuscript. Please note that it should not be uploaded as file type “Other”, which is not viewable by the reviewers.

2) Thank you for providing us with the Data Availability Statement. Please provide us with a direct link to access the dataset.

**Reviewers' Comments:**Reviewer's Responses to Questions

**Part I - Summary**

Reviewer #2: The manuscript reports antiplasmodial properties of quinoxalines and alludes to its mechanism of action. The authors have responded well to comments from the previous submission.

Reviewer #3: (No Response)

Reviewer #4: This manuscript reports on the activities of anti-schistosomal quinoxaline molecules repurposed to target malaria. This is the first time I have reviewed this manuscript and note that it has been previously reviewed and revised. The authors determine the activity of “compound 22” in standard asexual growth assays and then demonstrate increased potency of analogues “31” and “33”. They demonstrate that all have similar activites against a range of different parasite isolates from different geographical (and drug sensistivity) backgrounds. The research team behind this work are renowned for their previous groundbreaking studies into determining antimalarial molecule mode of action/mechanism of resistance and they employ their well-validated pipeline to probe the quinoxalines. Through drug pressure to compound 22, modest (but statistically significant) resistance was generated and through genome sequencing, PfQRP1 mutations were implicated. Two lines possessing mutations in this gene were tested and they recapitulated the resistance phenotype. However, as PfQRP1 is non-essential, the authors conclude that this is likely a mechanism of resistance rather than the molecular target. Cleverly, the authors also generated a PfQRP1 mutant with an inactive catalytic triad that was substantially more resistant to compound 22 and 31, but not 33. Having detected copy number variants for the flippase gene pfatp2 the authors also note that this gene also contributes to resistance.

Reviewing this manuscript at this late stage, previous reviewers have highlighted suggested improvements that the authors have met. This study is thorough and proposes a new class of molecules that are dual active against two significant parasites.

**Part II – Major Issues: Key Experiments Required for Acceptance**

Reviewer #2: (No Response)

Reviewer #3: The authors did not address a major concern about pfatp2 being the target of compounds. There is no biochemical assay or genetic validation. Failure to obtain a KO is not evidence of essentiality. They refer to work by Dr Lehane in biorxiv but I was unable to find the preprint.

Reviewer #4: No major revisions required.

**Part III – Minor Issues: Editorial and Data Presentation Modifications**

Reviewer #2: (No Response)

Reviewer #3: (No Response)

Reviewer #4: The manuscript is thorough and research is well conducted. Noting the comments of previous reviewers, it is my opinion that the authors have addressed previously raised comments and I do not have any major concerns myself. There is always more characterization that could be performed – such as in vivo efficacy in a rodent model (or humanized mice) but I believe that this would be out of the scope of the manuscript and could form the basis of future work if this compound was progressed with a view to further development. From a cell biology perspective, given that other recently reported antimalarials target lipid transfer (e.g PfSTART), it would be interesting to study the effects of the quinoxalines on cellular morphology (i.e. membrane transfer, formation of the amoeboid form etc). However these also could form another manuscript.

PLOS authors have the option to publish the peer review history of their article (what does this mean? ). If published, this will include your full peer review and any attached files.

**Do you want your identity to be public for this peer review?** For information about this choice, including consent withdrawal, please see our Privacy Policy .

Reviewer #2: No

Reviewer #3: No

Reviewer #4: **Yes: ** Michael Delves

---

## [Editor Report · Decision Letter 2]

14 Jan 2025

Dear Professor Lee,

We are pleased to inform you that your manuscript 'Quinoxaline-Based Anti-Schistosomal Compounds Have Potent Anti-Plasmodial Activity' has been provisionally accepted for publication in PLOS Pathogens.

Best regards,

Richard J. Martin, BVSc, PhD, DSc, DipECVPT, FRCVS

Guest Editor

PLOS Pathogens

Dominique Soldati-Favre

Section Editor

PLOS Pathogens

Sumita Bhaduri-McIntosh

Editor-in-Chief

PLOS Pathogens

orcid.org/0000-0003-2946-9497

Michael Malim

Editor-in-Chief

PLOS Pathogens

orcid.org/0000-0002-7699-2064

In the opinion of this Editor, the manuscript is now acceptable for publication.